# Outcomes and Adverse Events in Patients with Cancer after Diagnosis of Immunotherapy-Associated Diabetes Mellitus: A Retrospective Cohort Study

**DOI:** 10.3390/cancers16091663

**Published:** 2024-04-25

**Authors:** Eva Duvalyan, Sam Brondfield, Robert J. Rushakoff, Mark S. Anderson, Zoe Quandt

**Affiliations:** 1School of Medicine, University of California San Francisco, San Francisco, CA 94143, USA; eva.duvalyan@ucsf.edu; 2Division of Hematology and Oncology, Department of Medicine, University of California San Francisco, San Francisco, CA 94143, USA; sam.brondfield@ucsf.edu; 3Division of Endocrinology and Metabolism, Department of Medicine, University of California San Francisco, San Francisco, CA 94143, USA; robert.rushakoff@ucsf.edu (R.J.R.); mark.anderson@ucsf.edu (M.S.A.); 4Diabetes Center, University of California San Francisco, San Francisco, CA 94143, USA

**Keywords:** checkpoint blockade, diabetes mellitus, immune tolerance, immune toxicity, autoimmunity, immune monitoring, metabolism

## Abstract

**Simple Summary:**

Checkpoint-inhibitor-associated diabetes mellitus is a severely morbid immune-related adverse event that occurs from immunotherapy treatment during cancer therapy. Many patients and clinicians opt to discontinue therapy after this adverse event occurs while others choose to resume treatment. There are currently no data to inform this decision in terms of risks and benefits to the patient. This study aims to determine future risk for immune-related adverse events and cancer outcomes in patients who make this decision after experiencing diabetes mellitus.

**Abstract:**

Immune checkpoint inhibitor (CPI)-induced diabetes mellitus (CPI-DM) is a rare immune-related adverse event (irAE). Patients and providers fear that continuing CPIs puts patients at risk for additional irAEs and thus may discontinue therapy. Currently, there are little data to inform this decision. Therefore, this study aims to elucidate whether discontinuing CPIs after diagnosis of CPI-DM impacts the development of future irAEs and cancer outcomes such as progression and death. Patients who developed CPI-DM during cancer treatment at UCSF from 1 July 2015 to 5 July 2023 were analyzed for cancer outcomes and irAE development. Fisher’s exact tests, Student *t*-tests, Kaplan–Meier methods, and Cox regression were used as appropriate. Of the 43 patients with CPI-DM, 20 (47%) resumed CPIs within 90 days of the irAE, 4 (9%) patients restarted after 90 days, and 19 (44%) patients never restarted. Subsequent irAEs were diagnosed in 9 of 24 (38%) who resumed CPIs and 3 of 19 (16%) who discontinued CPIs (*p* = 0.17). There was no significant difference in death (*p* = 0.74) or cancer progression (*p* = 0.55) between these two groups. While our single-institution study did not show worse cancer outcomes after discontinuing CPIs, many variables can impact outcomes, which our study was not adequately powered to evaluate. A nuanced approach is needed to decide whether to continue CPI treatment after a severe irAE like CPI-DM.

## 1. Introduction

Immune checkpoint blockade is a rapidly expanding modality of cancer treatment and continues to gain approval for more cancer indications. This class of immunotherapy encompasses drugs with mechanisms that include antagonizing programmed death-ligand 1 (PD-L1), programmed cell death protein 1 (PD-1), cytotoxic T-lymphocyte-associated protein 4 (CTLA-4), and, most recently, lymphocyte-activation gene 3 (LAG-3) [1,2]. The blockade of these checkpoints on cancer cells induces the immune system recognition of the cells as foreign and prompts anticancer activity, thus leading to the eventual death of malignant cells. However, this mechanism also poses a risk to healthy cells, as breaks in immune tolerance lead to autoimmune and autoinflammatory side effects, which are known as immune-related adverse events (irAEs), which can occur in most organ systems [3].

Checkpoint inhibitor diabetes mellitus (CPI-DM) is a severe and highly morbid irAE caused by the collateral damage from autoimmune activation targeting pancreatic beta cells. It occurs in 0.2–1.9% of patients using CPIs, is predominantly in those exposed to PD-1 and PD-L1 inhibitors, and it is not seen in CTLA-4 inhibitor monotherapy [4]. Phase 2 and 3 clinical trials for the only LAG-3 inhibitor available in the US did not list CPI-DM as an irAE that occurred in ≥1% of study participants [5]. The autoantibodies usually present in type 1 diabetes (T1DM) such as GAD65, IA-2 (also known as ICA-512), insulin auto-antibodies, and ZnT8 are positive in some cases but not all [6]. While the mechanism is unknown currently, this suggests that it differs from traditional T1DM. For example, in studies where GAD65 is measured, less than half of patients with CPI-DM show GAD65 positivity [4,7,8].

In nearly all cases of CPI-DM, presentation involves a precipitous and significant increase in blood glucose levels, with low or undetectable C-peptide levels present in 73–91% of cases, and positive autoantibodies associated with type 1 diabetes such as GAD65 seen in 48–52% of patients [9,10]. In 59–71% of cases, depending on the study, patients will develop potentially life-threatening diabetic ketoacidosis (DKA), which requires hospitalization to initiate insulin therapy and replete fluids [8,11,12]. Diagnosis is often made with standard lab values such as blood glucose, urine, or serum ketones, and blood pH and treatment is similar to T1DM with short and long-acting insulin. Auto-antigen, auto-antibody, insulin, and C-peptide levels are not routinely measured as they are not necessary to treat the patient, although they can sometimes help to confirm the diagnosis. The discontinuation of CPI therapy almost never reverses CPI-DM, and treatment requires lifelong insulin therapy [3,4,8,13]. Due to the dramatic consequences and often severe presentation of this irAE, patients and physicians may be fearful of resuming CPI treatment due to the risk of additional irAEs, while others may continue treatment due to the irreversible and medically manageable nature of CPI-DM. These decisions have the potential to impact the patient’s cancer outcomes and possibly put the patient at risk for future morbidities.

Several studies have described the clinical syndrome of CPI-DM and theorized its pathophysiologic mechanism related to checkpoint blockade but none, to our knowledge, have explored the implication of the clinical decision of either resuming or discontinuing CPI treatment following the diagnosis of CPI-DM. Thus, there has been no guidance to help decide whether to discontinue or resume CPI therapy. To date, this decision is often made by assessing a variety of factors, including the cancer’s response to treatment and patient preference. To provide guidance when faced with similar situations, this study evaluated the incidence of additional irAEs and cancer outcomes in patients who have either resumed or discontinued CPI therapy after developing CPI-DM.

## 2. Materials and Methods

This is a single-institution retrospective cohort study including patients who developed CPI-DM between 1 July 2015 to 5 July 2023 while being treated with a PD-1 or PD-L1 inhibitor alone or in combination with other cancer therapies. CPI-DM is defined as a loss of insulin production, as defined by low C-peptide and presentation in DKA [8,14,15]. All of the patients in the study required prolonged basal–bolus insulin administration. The positive autoantibodies typically present in T1DM were not required to be included in the study. This study was approved by the UCSF Institutional Review Board.

Study endpoints included the development of new irAEs and cancer outcomes after CPI-DM diagnosis in those who discontinued CPIs compared to those who resumed. Patients were categorized into three groups: those who resumed CPIs within 90 days after DM diagnosis (prompt restart), those who resumed CPIs after 90 days (delayed restart), and those who permanently discontinued CPI therapy (Figure 1). New irAEs were defined as symptoms or syndromes that were either biopsy-proven or clinically determined to be due to CPI therapy by the treating physician(s) after ruling out other attributing causes. Progression was determined in accordance with Response Evaluation Criteria in Solid Tumors (RECIST) guidelines [16]. Patient data were collected until either death, loss to institution-specific follow-up, or 9 October 2023, whichever came first.

Baseline demographic data, including age, sex, type of cancer, prior lines of cancer therapy, prior irAEs, and personal or family history of DM, as well as other autoimmune conditions, were collected (Table 1). The date of CPI-DM diagnosis was recorded along with presenting symptoms, presence of diabetic ketoacidosis, and insulin requirement. After CPI-DM diagnosis, new irAEs were documented for all patients, including the respective irAE treatments required. Outcome data were collected for all patients, including response at CPI-DM diagnosis, progression, and death.

Additional analysis was performed comparing the incidence rate of new irAEs based on the time the patient was on and off CPI treatment (Figure 2). On-treatment time was defined as the time from CPI-DM diagnosis to the date of the patients last CPI cycle in those who resumed CPI. Off-treatment time began on the date of the CPI-DM diagnosis, and it was calculated with three different end dates to examine the effect of delayed irAEs. The first approach used the time from CPI-DM diagnosis to either the date of death or last point of contact. The second approach limited the time off treatment post-CPI exposure to one year (365 days) maximum, and the third approach limited the time off treatment post-CPI exposure to six months (180 days) maximum. In patients who had a prolonged delay between CPI-DM diagnosis and resumption of CPI therapy, the delayed time to resumption was considered time off-treatment.

Baseline characteristics, new irAEs, and cancer outcomes were summarized using descriptive statistics, such as medians, ranges, counts, and percentages, and they were compared by CPI use after CPI-DM diagnosis using binomial tests, *t*-tests, and Fisher’s exact tests, as appropriate. The time-to-event data were summarized with Cox regression and log-rank tests, and they were visualized with Kaplan–Meier methods. Analyses were performed using the R statistical program version 4.3.1.

## 3. Results

Forty-three patients that developed CPI-DM during the study period were identified. Among those who died, the median follow-up time was 535 (51–2525) days, while all other patients were followed with at 1368 (96–2784) days.

### 3.1. Baseline Demographics

Of the 43 patients with CPI-DM, 19 (44%) discontinued CPIs after diagnosis, 20 (47%) had a prompt restart within 90 days, and 4 (9%) had a delayed CPI restart (Figure 1). Delays in restarting CPIs were either due to patient preference (*n* = 1) or disease progression (*n* = 3). Patients were treated for a variety of cancers, with melanoma being the most common (*n* = 11, 26%), followed by non-small cell lung cancer and breast cancer (both with *n* = 4, 9%). The most common CPIs used prior to CPI-DM onset were pembrolizumab (*n* = 23, 53%) and nivolumab (*n* = 14, 33%). A total of 40 patients (93%) received PD-1 or PD-L1 blockades, while the remaining 3 (7%) received a combination of PD-1 and CTLA-4 inhibitor. Furthermore, 9 (21%) patients had a history of pre-diabetes and 3 (7%) had controlled type 2 diabetes mellitus (T2DM). Patients with a history of pre-CPI prediabetes or diabetes were more likely to resume CPIs after CPI-DM diagnosis. In addition, 4 (9%) patients had a family history of type 1 diabetes mellitus (T1DM). Prior thyroid disease was present in 21 (49%) of patients. Moreover, 5 patients (12%) had autoimmune thyroid diseases diagnosed and 16 (37%) had already developed CPI-induced thyroid dysfunction. Dermatitis (*n* = 10, 23%), arthritis (*n* = 6, 14%), and colitis (*n* = 5, 12%) were among the most common non-endocrine irAEs that occurred prior to CPI-DM onset. Overall, only 28 (65%) of the patients had any irAEs before CPI-DM was diagnosed (Table 1). The time from the beginning to the last CPI therapy to diagnosis CPI-DM was not significantly associated with the decision to resume or discontinue CPI therapy (*p* = 0.078) (Figure 3A). Furthermore, 20 out of 24 patients (83%) resumed CPI therapy within the first three months after CPI-DM diagnosis (Figure 3B).

### 3.2. New irAEs Diagnosed after CPI-DM

Of the 43 patients with CPI-DM, 12 (28%) developed new irAEs after being diagnosed with CPI-DM, regardless of whether CPIs were resumed or not (Figure 1). Moreover, 7 (35%) patients who promptly resumed, 2 (50%) of those who delayed resuming (both of which occurred after resuming CPIs), and 3 (16%) of those who permanently stopped CPI had subsequent irAEs after being diagnosed with CPI-DM. Resuming CPI therapy was not associated with the development of new irAEs (38% resuming CPI and 16% permanently stopping CPI, *p* = 0.17). Factors that may be signs of increased autoimmunity such as prior irAEs and a family history of autoimmune disease did not predict new irAE development (Table 2). The development of new irAEs was not influenced by the time from CPI initiation to CPI-DM diagnosis (Figure 4). Among the patients who developed CPI-DM within 3, 3–6, 6–12, and over 12 months from starting CPI therapy, 3/13 (23%), 2/13 (15%), 7/14 (50%), and 0/3 (0%) developed a subsequent irAE, respectively (*p* = 0.17). This association with time to CPI-DM diagnosis remained non-significant regardless of whether the first exposure to CPI was used as the starting point or the start date of the CPI that the patient was on when CPI-DM diagnosis was considered (Kaplan–Meier, log rank *p* = 0.8 and 0.7, respectively).

### 3.3. Cancer Outcomes

When considering cancer outcomes, patients who had a delayed resumption of CPIs (>90 days after CPI-DM diagnosis) were grouped with those who discontinued CPIs as this was in line with the treating physician’s intent at CPI-DM diagnosis. The patient cancer response at CPI-DM diagnosis did not significantly impact whether the patients resumed CPI or stopped/delayed resuming them (*p* = 0.16). However, there was a tendency for more patients with PR or SD to promptly resume CPI therapy (16/20, 75%) compared to stopping or delaying CPI resumption (11/23, 48%) (*p* = 0.06).

Furthermore, 6 (32%) patients who stopped/delayed resuming CPIs and 7 (29%) patients who promptly resumed CPIs died during the study period (*p* = 0.74). Among the patients who died from non-cancer related causes, 3 (16%) had stopped/delayed CPIs and 1 (4%) had promptly resumed CPIs. The alternate causes of death included hip fracture, gastrointestinal stromal tumor progression (which was different from the initial malignancy requiring CPI use), aspiration pneumonia, and a surgical complication. When limiting the analysis to cancer-attributable deaths, 3 (7.7%) and 6 (15%) of the patients who stopped and resumed CPIs died from their original cancer (*p* = 0.71). Moreover, 8 (35%) patients who stopped/delayed CPIs and 9 (45%) patients who resumed them had a progression of their disease (*p* = 0.55). All of the patients who had a prolonged pause (>90 days) in restarting CPI therapy had either no evidence of disease, stable disease, or, in one subject, a partial response at CPI-DM diagnosis that converted to a complete response in the coming months.

The median time to death was 1139 days and 805 days in patients who stopped and resumed CPIs (*p* = 0.36), while the median time to progression was 566 and 492 days in both groups (*p* = 0.47), respectively (Table 3).

## 4. Discussion

In our review of 43 patients with CPI-DM, there was no difference in the incidence of new irAEs or an effect on mortality and progression between those that stopped and resumed CPIs.

Patients with CPI-DM remained at risk for subsequent irAEs both on and off CPI therapy. In our study, 38% of patients who resumed therapy and 16% who discontinued CPI developed new irAEs. Risk of irAEs after resuming CPI therapy in the context of CPI-DM may be slightly lower than resuming CPIs after other irAEs. In similar studies, 52–55% of patients develop new or recurrent irAEs when CPIs are rechallenged after any irAE and higher severity irAEs (like CPI-DM), which carry a non-significant but higher risk of additional irAEs upon CPI resumption compared to lower severity irAEs [17,18]. Importantly, unlike irAEs such as colitis, inflammatory arthritis, or dermatitis, CPI-DM is nearly always a permanent irAE that cannot recur or worsen during the CPI treatment course. Since CPI-DM typically has a longer range of time to presentation compared to other irAEs, the lower incidence of new irAEs in our study may be explained by the fact that many patients will have already experienced other irAEs that they are most susceptible to prior to the onset of CPI-DM [19,20]. This is reflected in our study with 65% of patients having experienced another irAE prior to developing CPI-DM. In summary, resuming CPIs does not seem to put patients at additional risk compared to having developed other irAEs, but further study is needed to determine if this risk is higher than patients who have continued on therapy without a history of severe irAE.

An important finding in this study is the confirmation that patients remain at risk for irAEs after discontinuation of CPIs, which is consistent with a prolonged alteration in immune tolerance. The occurrence of delayed irAEs is mechanistically supported by Phase I pharmacodynamic studies showing PD-1 receptor occupancy remains at a plateau of 72% for ≥ 57 days, despite the serum half-life of the drug being 12–20 days [21]. A more recent study on the receptor occupancy (RO) of PD-1 drugs showed a two-fold reduction in antibody binding capacity in patients exposed to CPIs within the last 14 weeks, presumably due to interfering RO from prior anti-PD1 drugs [22]. Furthermore, a study on delayed irAEs after CPI discontinuation showed a median time to delayed irAE of 6 months in a cohort of 23 patients [23]. Our sensitivity analysis showed that the rate of irAEs does decrease with the passage of time after CPI discontinuation and that patients are statistically less likely to obtain new irAEs somewhere between 6–12 months after their last CPI exposure. This highlights the persistent effect of CPIs on immune tolerance and supports the need to remain vigilant for irAEs post-CPI cessation for at least 6 months (but possibly longer).

This study did not reveal a significant effect on cancer progression and cancer-related mortality when discontinuing CPIs. The trend toward significantly more patients with PR or SD promptly resuming CPI therapy (*p* = 0.06) suggests that cancer providers may have been more likely to continue treatment if patients seemed to have tenuous control over their cancer. This is clinically reasonable and reflects the complexity of treatment decisions made during cancer care that includes a careful analysis of risks, benefits, the current state of the patient’s cancer control, and their functional status.

A limitation in this study is the small sample size, which has possibly prevented the detection of significant differences in the outcomes and incidence of irAEs. Additionally, there are likely confounding factors surrounding the cancer outcomes that have not been captured in this study, including cancer grade, cytogenetics, stage at diagnosis, and patient performance status, among others. Furthermore, the heterogeneity of cancer types in our relatively small patient population may mask the inherent differences in the expected response to immunotherapy and expected rates of progression and death. Thus, cancer characteristics could be more meaningful, and the differences in cancer outcomes could be more interpretable when examining large cohorts with more patients with a single tumor type. Therefore, future research should focus on recruiting a larger cohort of patients or combining data across institutions to increase study power. Additionally, other endpoints should be explored, such as quality-adjusted life year analyses to better understand patient perspectives in addition to clinical endpoints. The authors of this study hope to update this article when more patients are received with CPI-DM at the institution where this study was conducted.

Immunotherapy has been an important recent discovery that has provided a major tool toward fighting cancer. It offers life-saving treatment for patients who would otherwise have no options. However, it is crucial to continue expanding our understanding of the irAEs associated with this therapy so we may better serve our patients in mitigating them and making treatment more tolerable [24,25]. As such, both future clinical, basic science, and translational research should work toward understanding irAE mechanisms and providing treatment or prevention strategies [26,27].

## 5. Conclusions

In conclusion, this study demonstrated no significant difference in the risk of additional irAEs and progression or death for patients who decide to stop or continue CPI therapy after CPI-DM. However, this remains a challenging clinical decision and further research is needed to provide clearer guidance to clinicians when making individualized decisions for their patients. Nevertheless, these data may provide clinicians and patients with an additional piece of information when considering the difficult decision of whether to resume CPI therapy. Most importantly, providers should counsel that discontinuing CPIs may not prevent all additional irAEs, especially in the immediate three-month period after the last dose.

## Figures and Tables

**Figure 1 cancers-16-01663-f001:**
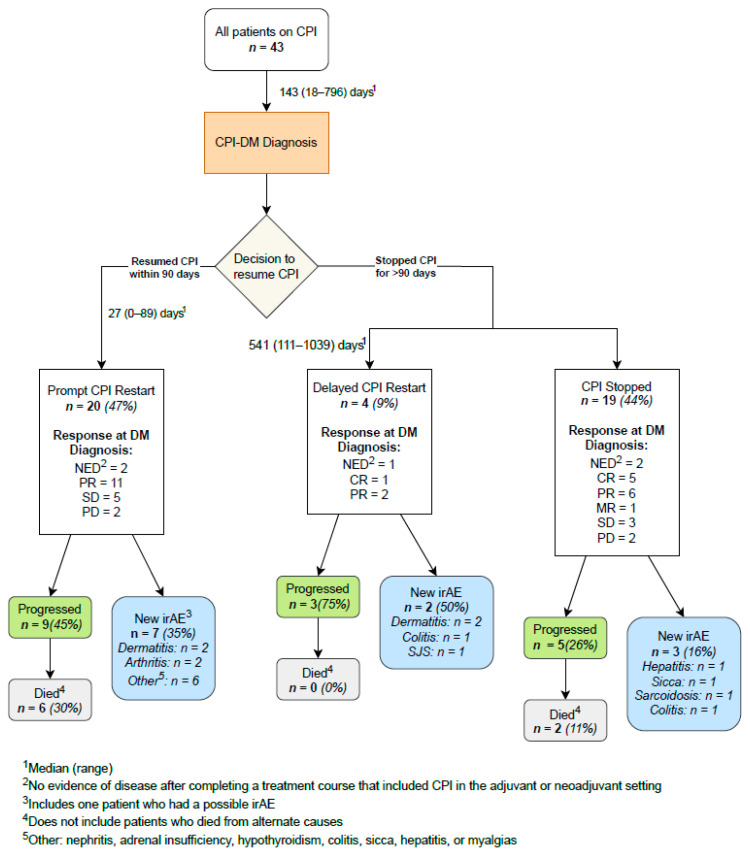
Flow chart depicting the clinical course of patients who continued on immune checkpoint inhibitors and who stopped immune checkpoint inhibitors at the time of immune checkpoint inhibitor diabetes.

**Figure 2 cancers-16-01663-f002:**
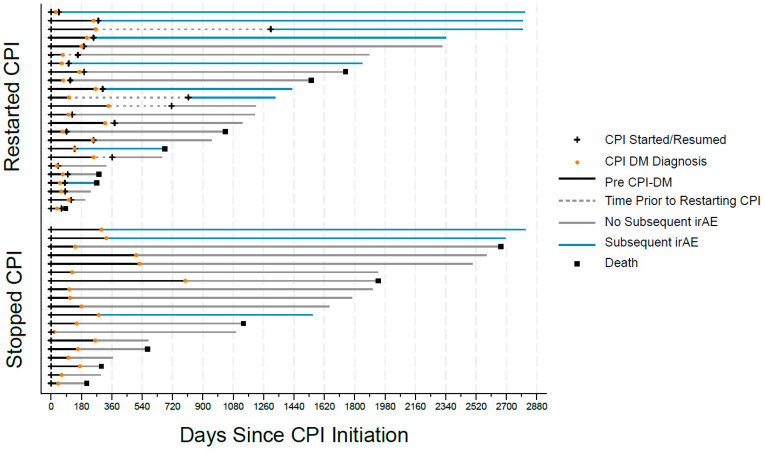
Timeline of the treatment course from the initiation of immune checkpoint inhibitors to diabetes diagnosis, as well as the subsequent treatment and new onset of immune-related adverse events, which were split by those who restarted immune checkpoint inhibitors at some point after diabetes diagnosis and those that permanently stopped immune checkpoint inhibitor therapy.

**Figure 3 cancers-16-01663-f003:**
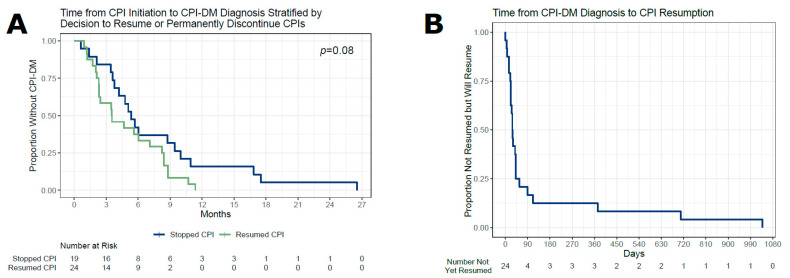
(**A**) Of the 43 patients who developed CPI-DM, 24 subjects resumed CPI therapy at some point and 19 did not. The likelihood of resuming CPI therapy after CPI-DM diagnosis did not differ by the length of exposure to the CPI prior to CPI-DM diagnosis (*p* = 0.078). (**B**) Of the 24 patients resuming CPIs after CPI-DM diagnosis, 20 did so within the first 90 days after CPI-DM diagnosis. The remaining 4 patients resumed CPI within 3 years.

**Figure 4 cancers-16-01663-f004:**
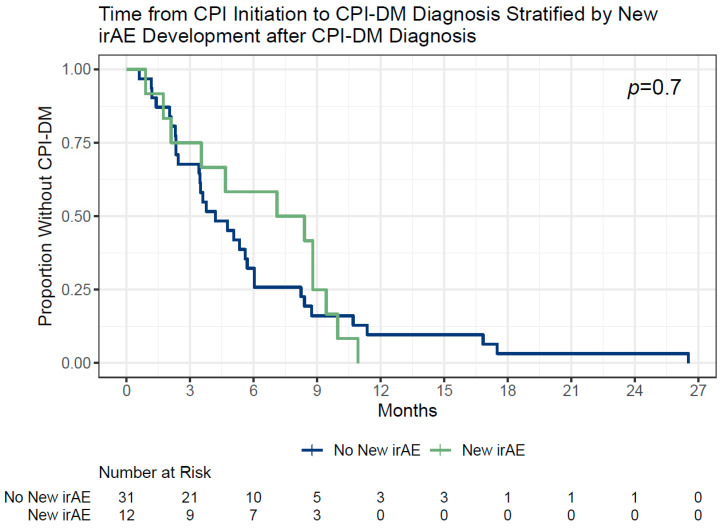
Of the 43 patients who developed CPI-DM, 12 subjects developed a new irAE after CPI-DM diagnosis and 31 did not. The risk of subsequent irAE after CPI-DM diagnosis did not differ by the length of exposure to the CPI prior to CPI-DM diagnosis (*p* = 0.7).

**Table 1 cancers-16-01663-t001:** Patient demographics.

	Total	Delayed CPI Restart ^1^	Prompt CPI Restart ^2^	Stopped CPI ^3^	*p*-Value ^4^
Number of Patients	43 (100)	4 (9)	20 (47)	19 (44)	0.54
Sex					
Male	25 (58)	1 (25)	14 (70)	10 (53)	0.55
Female	18 (42)	3 (75)	6 (30)	9 (47)	
Age	62.6 (37–87)	71.8 (67–77)	59.6 (37–80)	63.8 (43–87)	0.53
Average BMI	24.7 (16–42.6)	24.6 (17.1–29.5)	24.9 (16–42.6)	24.6 (20.7–31.7)	0.84
Race/Ethnicity					
White	28 (65)	2 (50)	12 (60)	14 (74)	0.35
Non-White ^5^	15 (35)	2 (50)	8 (40)	5 (26)	
Type of Cancer					
Melanoma	11 (26)	1 (25)	4 (20)	6 (32)	0.49
Non-Melanoma ^6^	32 (74)	3 (75)	16 (80)	13 (68)	
CPI Class					
PD-1/PD-L1 Inhibitor ^7^	40 (93)	4 (100)	19 (95)	17 (89)	0.58
PD-1 and CTLA-4 Combination ^8^	3 (7)	0	1 (5)	2 (11)	
Adjuvant CPI	9 (21)	2 (50)	2 (10)	5 (26)	0.48
Prior Treatment	38 (88)	4 (100)	19 (95)	15 (79)	0.15
Surgery	24 (56)	3 (75)	10 (50)	11 (58)	
Radiation	16 (37)	2 (50)	7 (35)	7 (37)	
Conventional Chemotherapy	17 (40)	3 (75)	7 (35)	7 (37)	
Targeted Therapy	2 (5)	0	2 (10)	0	
Other Therapy ^9^	6 (14)	0	4 (20)	2 (11)	
Any Prior irAE	28 (65)	3 (75)	12 (60)	13 (68)	0.76
Prior Endocrine irAEs					
CPI Thyroid Dysfunction	15 (35)	0	10 (50)	5 (26)	0.6
CPI Hypopituitarism	4 (9)	0	2 (10)	2 (11)	
Prior Non-Endocrine irAEs	23 (53)	3 (75)	8 (40)	12 (63)	0.36
Dermatitis	10 (23)	1 (25)	3 (15)	6 (32)	
Colitis	5 (12)	0	2 (10)	3 (16)	
Arthritis	6 (14)	2 (50)	1 (5)	3 (16)	
Hepatitis	3 (7)	0	1 (5)	2 (11)	
Sicca	2 (5)	0	1 (5)	1 (5)	
Pancreatitis	3 (7)	0	1 (5)	2 (11)	
Other Prior irAE ^10^	11 (26)	2 (50)	4 (20)	5 (26)	
History of Diabetes	12 (28)	1 (25)	9 (45)	2 (11)	0.04
Prediabetes	9 (21)	1 (25)	6 (30)	2 (11)	
T2DM	3 (7)	0	3 (15)	0	
History of Hypothyroidism *	11 (26)	1 (25)	3 (15)	7 (37)	0.17
Hashimoto’s Disease	7 (16)	0	2 (10)	5 (26)	
RT-Induced	1 (2)	1 (25)	0	0	
Ablation	1 (2)	0	0	1 (5)	
Thyroidectomy	1 (2)	0	0	1 (5)	
History of Autoimmune or Atopic Disease	15 (35)	0	5 (25)	10 (53)	
Allergies	7 (16)	0	3 (15)	4 (21)	
Asthma	3 (7)	0	2 (10)	1 (5)	
Rheumatoid Arthritis	1 (2)	0	0	1 (5)	
Graves’ Disease	1 (2)	0	0	1 (5)	
Other Autoimmune Disease	8 (19)	0	3 (15)	5 (26)	
Family History of DM *	18 (42)	2 (50)	11 (55)	5 (26)	0.12
First Degree Relative	10 (23)	0	6 (30)	4 (21)	
Second Degree Relative	5 (12)	1 (25)	4 (20)	0	
Third Degree Relative	4 (9)	1 (25)	2 (10)	1 (5)	
Family History of T1DM	4 (9)	2 (50)	0	2 (11)	0.53
Family History of Autoimmune Diseases ^11^	10 (23)	2 (50)	5 (25)	3 (16)	0.47

*Note:* All values are n (%), except age and average BMI (which are represented as mean (range)). BMI = basal metabolic index, CPI = checkpoint inhibitor, PD-1 = programmed cell death protein 1, PD-L1 = programmed death-ligand 1, CTLA-4 = cytotoxic T-lymphocyte-associated protein 4, irAE = immune-related adverse event, DM = diabetes mellitus, T2DM = type 2 diabetes mellitus, RT = radiation therapy, and T1DM = type 1 diabetes mellitus. ^1^ Resumed CPI therapy > 90 days after CPI-DM diagnosis. ^2^ Resumed CPI therapy ≤ 90 days after CPI-DM diagnosis. ^3^ Among patients who discontinued CPI at the time of diabetes diagnosis, four had already completed therapy or recently discontinued their CPI, and one had no evidence of disease that contributed to the decision to stop therapy. ^4^ The *p*-values throughout table were calculated by comparing patients who had a delayed or prompt CPI restart vs. those who stopped CPI therapy. ^5^ Non-white includes Hispanic, Asian, Pacific Islander, American-Indian Alaska Native, Filipino, or Other Non-Hispanic (as stated in the patient’s electronic medical record). ^6^ Non-melanoma includes head and neck squamous cell carcinoma, non-small cell lung cancer, breast cancer, renal cell carcinoma, Merkel cell carcinoma, cholangiocarcinoma, glioblastoma, urothelial cancer, gastroesophageal junction adenocarcinoma, gastric adenocarcinoma, prostate cancer, papillary thyroid cancer, pancreatic adenocarcinoma, mesothelioma, clear cell meningioma, endometrial adenocarcinoma, bladder cancer, neuroendocrine lung cancer, neuroendocrine colon cancer, and neuroendocrine pancreatic cancer. ^7^ PD-1 inhibitors include pembrolizumab and nivolumab. PD-L1 inhibitors include atezolizumab and durvalumab. ^8^ Includes ipilimumab and nivolumab. ^9^ Other therapies includes hormonal treatment, radioactive iodine therapy, octreotide, topical fluorouracil, or BCG with interferon. ^10^ Other prior irAEs include myocarditis, pneumonitis, nephritis, uveitis, vitiligo, NMDA+ receptor encephalitis, myasthenia gravis, and sarcoidosis. ^11^ Family history of autoimmune disease includes rheumatoid arthritis, unspecified thyroid disease, fibromyalgia, juvenile rheumatoid arthritis, multiple sclerosis, asthma, hypothyroidism, and Graves’ disease. All of the family history of autoimmune conditions were among first-degree relatives. * The *p*-value was calculated by comparing the presence of any subtype in this category vs. none.

**Table 2 cancers-16-01663-t002:** New irAE development after CPI-DM diagnosis.

	No New irAE	New irAE	*p*-Value
Number of Patients	31 (72)	12 (28)	
CPI Re-exposed and Class if Resumed ^1^	15 (48)	9 (75)	0.17
PD-1 Inhibitor ^2^	15 (48)	7 (58)	
PD-L1 Inhibitor ^3^	0	2 (17)	
CPI Discontinued	16 (52)	3 (25)	
History of Prior irAE	18 (58)	10 (83)	0.16
Personal History of Autoimmunity ^4^	11 (35)	4 (33)	1
Family History of DM	13 (42)	5 (42)	0.16
Family History of Autoimmunity ^5^	6 (19)	4 (33)	0.08

*Note:* All values are *n* (%). irAE = immune-related adverse event, CPI = checkpoint inhibitor, DM = diabetes mellitus, PD-1 = programmed cell death protein 1, and PD-L1 = programmed death-ligand 1. ^1^ Includes both patients who promptly resumed CPI therapy and those who delayed resumption. ^2^ Includes pembrolizumab and nivolumab. ^3^ Includes durvalumab and atezolizumab. ^4^ The personal history of autoimmune conditions, which includes rheumatoid arthritis, Graves’ disease, multinodular goiter, allergies, and asthma. ^5^ The autoimmune conditions present within family members, which includes rheumatoid arthritis, fibromyalgia, juvenile rheumatoid arthritis, thyroid disease (including Graves’ Disease and hypothyroidism), multiple sclerosis, and asthma.

**Table 3 cancers-16-01663-t003:** Cancer outcomes.

	Prompt CPI Restart ^1^	CPI Stopped or Delayed Restart ^2^	*p*-Value
Status at DM Diagnosis			
CR	2 (10)	9 (39)	0.74 ^3^
PR	11 (55)	8 (35)	
SD	5 (25)	3 (13)	
MR	0	1 (4)	
PD	2 (10)	2 (9)	
Progression			
Progression	9 (45)	8 (35)	0.55
No Progression	11 (55)	15 (65)	
Time to Progression (days)	492 (75–1193)	566 (89–1281)	0.47
Vital Status			
Alive	13 (65)	17 (74)	0.74
Dead	7 (35)	6 (26)	
Time to Death (days)	805.4 (87–1746)	1138.5 (212–2668)	0.36
Alternate Cause of Death	1 (5)	3 (13)	

*Note:* DM = diabetes mellitus, CR = complete response, PR = partial response, SD = stable disease, MR = mixed response, and PD = progressive disease. ^1^ Resumed CPI therapy ≤ 90 days after CPI-DM diagnosis. ^2^ Resumed CPI therapy > 90 days after CPI-DM diagnosis. ^3^ The *p*-value was calculated by grouping patients with CR and PR together versus those with SD, MR, and PD at CPI-DM diagnosis.

## Data Availability

The data presented in this study are available on request from the corresponding authors.

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
