# Peer review of "Outcomes and Adverse Events in Patients with Cancer after Diagnosis of Immunotherapy-Associated Diabetes Mellitus: A Retrospective Cohort Study"

_cancers, 2024, doi:10.3390/cancers16091663_

Round 1

Reviewer 1 Report

Comments and Suggestions for Authors

The study reported elucidation statistically whether discontinuing CPIs after diagnosis of CPI-DM would impact development of future irAEs and cancer outcomes. On that, more samples should be recruited for the study and Patients' background information, other than cancer stage should be elaborated. On top of that, would any other supplementary biomarkers to be used to provide supportive evidence to clarify the point at issue. Finally, more updated references should be cited. 

Additional comments on the manuscript as follows: 1. How did Immune checkpoint inhibitor (CPI)-induce diabetes mellitus (CPI-DM)? Since this is a rare incidence, what is the main question addressed by the research? 2. What is the significance of the present findings that can help address the problem at issue? 3. The immune system can be strengthened to fight against diseases but how did the immune inhibitor lead to the adverse effects? But what does the study add to the subject area compared with the existing literature? 4. What other experimental biomarkers such as insulin level can be used to provide supportive experimental evidence to address the main question of the present study? The methodology should include an appropriate control and more samples be recruited for the present study. P-value should be much less than 0.5 to show the significant difference among the controls and samples; thus, the statistical analysis of data shown in Tables and figures can be improved with sample size. The conclusion is not quite consistent with the present findings. 5. What parts of the findings are more relevant to cancer treatment or management of health? More updated references on inhibitors and inducers of the immune system should be cited. Comments on the Quality of English Language

nil

Author Response

Point-by-point response to Comments and Suggestions for Authors

Comments 1: The study reported elucidation statistically whether discontinuing CPIs after diagnosis of CPI-DM would impact development of future irAEs and cancer outcomes. On that, more samples should be recruited for the study and Patients' background information, other than cancer stage should be elaborated. On top of that, would any other supplementary biomarkers to be used to provide supportive evidence to clarify the point at issue. Finally, more updated references should be cited.

Response 1:

Thank you for the insightful comments! We agree that one of the limitations of this study is the relatively small sample size. As this is a very rare adverse event, it was challenging to capture a larger number of patients for this study. Future studies should consider combining patient cohorts across institutions to increase the power to detect statistically significant differences. This is already stated in our limitations, but we have added that an update to the article once more patients who experience CPI-DM are discovered at our institution would be helpful in creating a more robust conclusion (page 11, paragraph 4, lines 271-273).

We agree that including a variety of factors that influence prognosis such as stage, grade, and cytogenetics would be helpful. However, due to the small sample size and wide variety of cancers included in this study, information on cancer characteristics would be unlikely to be meaningful in our study since there would be too few patients in each category of cancer type and severity. In larger studies and perhaps those that focus on cancer within a single organ system or cell type, we agree that having individual cancer characteristics would be extremely helpful. We will add this distinction to our limitations paragraph (page 11 paragraph 4, lines 263-267).

An additional biomarker often used to categorize CPI-DM is C-peptide levels which correlate with the amount of endogenous insulin the body is producing. To provide more clarity on our methodology regarding this biomarker, we have included a remark on the C-peptide levels stating that patients either presented in diabetic ketoacidosis or had low C-peptide levels at diagnosis in our study (page 3, paragraph 1, lines 89-92).

Thank you for bringing to our attention that some of our citations are outdated. Our oldest citation was from a 2010 study and we have added an additional citation from a 2016 study to support our writing in this area (page 11, paragraph 2, lines 242-244). We have also added 11 more references from recent studies in our introduction and discussion sections.

Comments 2: How did Immune checkpoint inhibitor (CPI)-induce diabetes mellitus (CPI-DM)? Since this is a rare incidence, what is the main question addressed by the research?

Response 2: Thank you for the question. The aim of our research is to determine if the decision to discontinue or resume CPIs after developing CPI-DM in patients with cancer is correlated with death, progression, or the development of future irAEs. We have modified the wording in the abstract (page 1, paragraph 2, lines 23-24) and can direct the reviewers to the location of the question in the introduction to help clarify this (main question is on page 2, paragraph 4, lines 82-84).

We hope we have understood the first question correctly but if it is asking how CPIs cause DM, then please refer to Response 4 below and how we have incorporated this into our manuscript.

Comments 3: What is the significance of the present findings that can help address the problem at issue?

Response 3: Thank you for asking us to clarify this. Our findings suggest no difference in death or progression in patients who stop or continue CPI treatment. Additionally, patients remain at risk for irAEs after discontinuing CPI treatment. We have emphasized this point in our conclusion to help drive it home to readers (page 12, paragraph 1, lines 287-291).

Comments 4: The immune system can be strengthened to fight against diseases but how did the immune inhibitor lead to the adverse effects? But what does the study add to the subject area compared with the existing literature?

Response 4: This is a great question. While the mechanism for CPI-DM is still not completely clear, we hypothesize that there is targeted T-cell response against insulin producing pancreatic beta cells. There have been suggestions that it likely differs from traditional type 1 diabetes since the usual auto-antibodies are not always present such as GAD65, IA-2, ICA-512, and ZnT8. We have added this information to the introduction (page 2, paragraph 2, lines 53-48).

Comments 5: What other experimental biomarkers such as insulin level can be used to provide supportive experimental evidence to address the main question of the present study? The methodology should include an appropriate control and more samples be recruited for the present study. P-value should be much less than 0.5 to show the significant difference among the controls and samples; thus, the statistical analysis of data shown in Tables and figures can be improved with sample size. The conclusion is not quite consistent with the present findings.

Response 5: We are also interested in considering the rate of irAE in this group of CPI-DM patients compared to patients with other irAEs or those without irAEs at a similar point in their treatment progress to better understand the impact of this diagnosis, but that is answering a different question. Our goal is to help our patients who have already been diagnosed with CPI-DM to better predict what is coming next.

We also ask for some clarification in the reviewer comment stating that the “p-value should be much less than 0.5”. Is this meant to say 0.05?

Within other projects, we are looking at biomarkers of CPI-DM and we look forward to sharing these findings when they are complete.

Comments 6: What parts of the findings are more relevant to cancer treatment or management of health? More updated references on inhibitors and inducers of the immune system should be cited.

Response 6: Thanks for bringing this to our attention. We have clarified the importance of this type of research to the field of medicine and science in the discussion section (page 11 paragraph 5, lines 274-280). Additionally, we have increased the number of references from 16 to 27 and supported our oldest citation (2010) with an additional newer one from 2016.

Reviewer 2 Report

Comments and Suggestions for Authors

This is a well-written, interesting paper on a clinically relevant issue. My sole suggestion is to present data of table 2 in a graph format with pertinent notes.  

This interesting clinical paper deals with a retrospective analysis of 43 patients with immune checkpoint-inhibitor (CPI) associated diabetes mellitus (DM), a condition that may follow immunotherapy treatment in cancer therapy. In particular, the authors report the effects of CPI discontinuation after DM diagnosis in terms of adverse effects and cancer outcome. CPI therapy was restarted at different time points after DM inception in most patients and permanently discontinued in the remaining patients. This 9-year survey showed no worse cancer outcome after discontinuation or any difference in death rate. This is the most relevant result of this study; an additional result is that CPI resumption  should not entail additional risk of adverse reactions.  Despite the relatively small number of patients examined with  this infrequent condition, their clinical data are well detailed, and conclusions are reasonably grounded. The references listed are pertinent. In my opinion, data in Table 2 should be presented as graphs with pertinent footnotes. Table 1 is quite complex and difficult to read. At least some of its data could also be presented as graphs. 

Author Response

Please find the word file attached

Reviewer 3 Report

Comments and Suggestions for Authors

The manuscript by Duvalyan et al reports a retrospective cohort study of the immunotherapy-associated diabetes mellitus

Limitations- that should be commented in the article

1. Number of cases = 43. Very low for 4 different checkpoint inhibitors which can be used alone or in combination

2. QALY analysis may help in taking the ethical decision on continuing the treatment.

3. These data allow to explore the mechanism for Type 1 diabetes mellitus allowing new avenues to understand the autoimmune process- No way back? why for? Lack of beta cell regeneration?

4. FIGURE 1- Please expand the letters, There should be a minimum size f he letters.

5. Patient segmentation? Combined therapies

6. Obviously more studies are needed- but this is a single center study. No way to implement a national retrospective study?

7. Consequently the statistical analysis is not relevant.

Author Response

Please find the word file attached

Round 2

Reviewer 3 Report

Comments and Suggestions for Authors

Authors modified the text and now introduction, methods, results and conclusions are useful in the context of checkpoint inhibitors induced diabetes